# Towards a Sustainable Society through Emerging Mobility Services: A Case of Autonomous Buses

**Kenichiro Chinen [1]**, **Yang Sun [2]**, **Mitsutaka Matsumoto [3],\*** and **Yoon-Young Chun [4]**

1   College of Business Administration, California State University, Sacramento, CA 95819, USA; chinen@csus.edu
2   College of Graduate Studies, California Northstate University, Elk Grove, CA 95757, USA; yang.sun@cnsu.edu
3   Advanced Manufacturing Research Institute, National Institute of Advanced Industrial Science and Technology (AIST), Tsukuba 305-8654, Japan
4   Research Institute of Science for Safety and Sustainability, National Institute of Advanced Industrial Science and Technology (AIST), Tsukuba 305-8569, Japan; yy.chun@aist.go.jp
\*   Correspondence: matsumoto-mi@aist.go.jp

**Abstract:** The topic of emerging mobility services has quickly received attention from scholars and media in recent years. Mobility services employing autonomous buses in transport systems is one such example. Mobility services using emerging technologies are expected to create social, economic, and environmental benefits. However, the potential benefits of emerging mobility services using autonomous technology will not be realized unless self-driving vehicles are accepted and used by many passengers. The recent worldwide pandemic caused us to recognize the benefits of autonomous technologies. This pretest-and-posttest designed research examines the predictors of willingness to ride autonomous buses in a closed environment. The results of this study indicate that a combination of factors, such as societal benefits, attitude and technology adoption, directly and indirectly influence an individual's acceptance of autonomous buses. This study finds that passengers' willingness to use emerging mobility services after a sample riding experience is higher than before having a sample riding experience.

**Keywords:** emerging mobility services; innovation; autonomous bus; consumer acceptance

## 1. Introduction

### 1.1. Research Background and Objectives

Shifts in thinking about society related to sustainability employing emerging technologies and innovation are changing the competitive landscape. To meet trends and challenges, companies, local governments, and countries need to change the way they think about products, emerging technologies, processes, and designs for the successful operation of a business. Innovative and creative thinking is what saved businesses out of the 2008 financial crisis and is what will bring them out of the current pandemic crises. Early movers in emerging mobility services will help develop a sustainable society that competitors will struggle to match.

In recent years, the topic of emerging mobility services has quickly received attention from scholars and media. Self-driving, by definition, requires no direct driver input to accelerate, brake, or steer vehicles while continually monitoring the roadway, according to the National Highway Traffic Safety Administration [1]. There are degrees of vehicle autonomy from Level 0 (no autonomous operation) to Level 5 (fully autonomous operation) [2]. Several studies have estimated the long-term implementation

of Level-5 autonomous vehicles [3,4], and one recently predicted that, by 2045, 50% of vehicle sales would comprise driverless cars [5]. Driven by optimistic predictions for self-driving, vehicle manufacturers have begun intensely targeting the development of automated vehicles. Since the introduction of autopilot features in 2014 [6], Tesla has upgraded its autopilot features with traffic-aware cruise control to match the speeds of vehicles in the surrounding traffic. Furthermore, Tesla's autosteer feature assists steering within clearly marked lanes [7]. Even traditional car manufacturers (e.g., General Motors, Mercedes-Benz, Nissan, Toyota, Audi, and Volvo) have prototyped driverless vehicles. Technology giant Apple's prototype self-driving car has been spotted on the street [8]. Sony, a company best-known for its consumer electronics and entertainment products (e.g., Walkman, PlayStation, CMOS image sensors, and Sony Pictures), has also joined the autonomous car business [9].

Passenger beliefs and general attitudes toward products are important information for manufacturers because their beliefs, perception, and attitudes affect their behaviors. This study explores passengers' willingness to use emerging mobility services, more specifically autonomous buses (ABs) in a closed environment. The purpose of this research is (1) to examine direct effects of passenger attitudes, perceived societal benefits, anxiety, and early technology on their willingness to use emerging mobility services for both before-ride experience and after-ride experience (e.g., Do passenger attitudes towards ABs positively influence their willingness to use emerging mobility services?) and (2) examine the roles of perceived benefits, passenger anxiety, passenger attitudes, and early technology adoption as mediators between latent variables and willingness to use emerging mobility services for both before-ride experience and after-ride experience (Do passenger perceived benefits towards ABs mediate negatively the negative relationship between anxiety and willingness to use emerging mobility services?). Further, we examine how passengers' willingness to use emerging mobility services changes before and after a sample riding experiences.

The existing research gap on passenger acceptance has been demonstrated based on imagined situations versus realistic passenger experiences. To the best of our knowledge, our research is the first pretest/posttest-designed study in the U.S. that investigated the underlying factors of the acceptance of semi-autonomous buses operated in a closed environment. A series of two surveys (i.e., Surveys 1 and 2) were administrated in February–March, 2019, and April–May, 2019, respectively. Survey 1 was the pretest survey collected prior to participants (business students at California State University, Sacramento) having the opportunity to experience a prototype semi-autonomous Olli bus service offered by Local Motors at California State University, Sacramento. Survey 2 was the post-test survey. The same group of 146 participants answered surveys for both pre and post AB ride experiences in Sacramento, a typical metropolitan region in the State of California, USA. To examine direct and indirect effects of passenger attitudes, perceived societal benefits, anxiety, and early technology on their willingness to use emerging mobility services, the structural equation model (SEM) framework is used to examine which types of transit passengers were most willing to use mobility services employing autonomous technologies. We conducted a confirmatory factor analysis to measure construct validity, followed by examining the model fit and hypotheses testing. To examine how the experience of riding ABs influences passengers' desires to use emerging mobility services, we used a nonparametric matched-pairs sign test.

### 1.2. Introducing Autonomous Bus (AB) as Emerging Mobility Services

The considerable progress of artificial intelligence (AI) over the last decade has paved the way for driverless driving technologies. Media reports of the increased efforts of many industrial players (e.g., Google autonomous vehicles on public streets) have brought more attention and have encouraged investments. Compared with other self-driving vehicles, ABs follow virtual, fixed short tracks at very low speeds. The role of bus drivers is thus replaced by safety-driven technologies to lower human driver-fatigue risks. These benefits are important because statistics show that most road accidents are caused by driver error related to fatigue, drug use, distractions, or recklessness [10]. Many researchers have shared cautious optimism about the impact of autonomous vehicles [11]. They reported that

autonomous cars helped create more steady traffic flows [12], which could drastically reduce fatal traffic accidents [13]. Unsurprisingly, we now can see the industry's increased efforts in developing autonomous technologies [14]. One example is the Chevy Cruise autonomous vehicle, which lacks a steering wheel, pedals, gear shifters, and a cockpit. Thus, there is no obvious way for a human to take control should anything go wrong [15]. Autonomous-vehicle technologies could also spawn new business models, such as car-sharing services [16], which could further ease traffic congestion [17]. Fuller [11] shared his cautious optimism regarding autonomous cars by highlighting their benefits and possibilities.

When assessing the current capability of autonomous-car technologies, it is, unfortunately, apparent that self-driving vehicles are said to be not ready for public deployment [18]. Development is progressing, but at a slower pace than anticipated [19]. The WAYMO vehicle originated as a driverless technology project of Google. WAYMO announced in October, 2020 that it would launch fully driverless rides to the public in Phoenix, Arizona during a global pandemic. One setting entails closed (e.g., geo-fenced) environments, wherein vehicles are limited to geographic areas (e.g., university campuses, airports, retirement villages) [20].

Uncertainty remains regarding when, how, or whether fully autonomous vehicles will become a part of our daily lives or a commodity that will complement or replace conventional transport systems [21]. We can nevertheless examine how people rate automated vehicles based on the state of the current developments. Thus, even now, we can imagine how those vehicles are likely to be harmonized with our daily mobility requirements. Any scenario that applies fully driverless vehicles will require us to reinvent the way we use automobiles and to further integrate personal mobility with public transportation to help solve urban-mobility challenges (e.g., safety, pollution, congestion) [21]. For example, ABs can deliver demand-responsive transport and may act as an alternative to a variety of traditional public-transport systems [22]. There are driverless automated trams and metros already in worldwide operations [23]. Nevertheless, ABs are distinguishable, because they can operate even in a mixed-traffic environment and are free from specialized infrastructures [22].

The year 2019 represented one of the major AB campaigns for closed environments. The basic requirement of providing a moving platform for commuters to (from) various stop (start) points is far surpassed with ABs. ABs (aka automated shuttles) have been well-tested and are operational in the US and worldwide. Local Motors, Inc. introduced its self-driving shuttle, known as Olli, to the campus of California State University, Sacramento [24]. The University of South Florida, in partnership with Coast Autonomous, hosted an autonomous-vehicle demonstration featuring a self-driving shuttle along a busy central walkway [25]. Texas Southern University hosted Houston's first autonomous shuttle, which transported students and staff around campus [26]. The Mcity Driverless Shuttle, manufactured by the French firm, NAVYA, ran on the University of Michigan campus from 2018 to 2019 [27]. Yutong Bus launched driverless buses in the "Intelligence Island" of Zhangjiang, Shanghai, China. Their ABs can drive themselves without human intervention under specified conditions [28].

The introduction of ABs in transport systems is expected to create social, economic, and environmental benefits, including increased traffic safety, improved accessibility in underserved communities, and reduced fuel consumption. Policymakers have the right to be skeptical of the idea of autonomous technology because of its impact on public-transit ridership [29]. For example, it is known that large-scale ABs or similar adaptations (e.g., autonomous taxis) can generate disadvantages to existing public transit systems. However, the transition to automated buses and taxis in more restricted modes could benefit the transportation industry and highly traveled community because of the increased productivity and reduced subsidies incurred [30]. In particular, small-capacity ABs without drivers could provide cost-effective alternatives for regional bus operations for which low-volume transit routes are often too expensive to run [31].

From an ecological perspective, the technical characteristics of autonomous vehicles, such as improved fuel economy, reduced traffic jams, and reduced parking infrastructure, could reduce energy use and related gas emissions [32]. On the other hand, consumer choices and their decision-making

processes related to adopting new technologies and services could have a significant potential to cancel out the benefits of the technological advances [32]. Recent environmental research has gravitated towards the former perspective, and the behavioral effect of autonomous vehicles on the environment remains mostly unexplored [33].

### 1.3. Importance of Understanding Consumer Acceptance of Emerging Mobility Services

People are aware of the environmental damage caused by combustion vehicles. Electric automobiles are expected to lead to significant reductions in worldwide $CO_2$, leading to a reduced dependency upon carbon-constrained energy in general [34]. ABs offer these advantages, in addition to potentially reducing greenhouse-gas emissions, air pollution, and vehicular noise. Morton et al. [35] examined related environmental concerns and reported the consequences of car use, self-reported human responsibilities, and willingness to pay to reduce emissions, learning that they are associated with positive attitudes towards electric vehicles.

In the US, 71% of respondents reported being afraid of riding in an automated car [36]. The most significant barrier to the uptake of autonomous vehicles may be attitudinal or emotional rather than technological [37]. The recent report from AAA (American Automobile Association) concluded that closing the perception and reality gap is key to the success of passenger acceptance toward autonomous vehicles [36]. To date, however, the status quo is not clearly understood [38,39]. Potential benefits of autonomous technology will thus not be realized unless self-driving vehicles are accepted and used by many passengers. Therefore, we investigated the issues of public acceptance or rejection of autonomous cars (i.e., ABs) for this study. This will enable us to better understand, predict, and assuage passenger concerns.

### 1.4. Studies on Emerging Mobility Services from the Consumer Perspective

Across the communities of psychological, sociological, information, and mobility services, theoretical frameworks have been proposed to explain the public acceptance of emerging mobility services and technologies. Still, psychological determinants are not well understood [39–41]. Additionally, many of the prior studies that used large national or cross-national datasets to ascertain people's opinions on various types of driverless concepts were based on imagined situations rather than those of realistic passenger experiences [42–46]. More recent studies, however, have examined the acceptance of passengers who had used autonomous vehicles in the recent past [22,39,40,47–49].

Giving passengers the opportunity to experience emerging technology could help reduce much of the public anxiety and lead to higher consumer acceptance. However, we could also argue that the reverse is true. Passenger experience with autonomous vehicles might increase public anxiety if something goes awry. In either case, a change in the degree of acceptability before and after experiencing autonomous transportation is expected [50]. The strength of the present study is that its respondents specifically experienced an AB. Thus, their direct experience can be assessed concerning future acceptance. It is hoped that a pretest–posttest-designed study will allow us to better capture and understand the factors influencing passenger decisions toward ABs.

The rest of the paper is organized as follows. First, it presents the theoretical background and research questions concerning ABs, thus formulating hypotheses for this research. This helps us clarify the potential factors that may influence users' acceptance of ABs. Second, the methodology used to conduct our investigation is explained. Third, we provide the results of the statistical analyses and hypothetical testing. In the last two sections, we summarize the research findings and conclude this research.

## 2. Conceptual Background and Hypotheses

The theory of planned behavior [51] supplies the theoretical foundation for the attitude construct of our study. The theory is based on the premise that, by analyzing the available knowledge, individuals make rational, reasonable decisions about participating in specific behaviors. A firm's business

performance, to a large extent, is influenced by the attitude of consumers who make the ultimate marketing decisions [52]. Consumer's beliefs and actions toward an object depend on how they perceive things. The market introduction of driverless cars is only meaningful if passengers are willing to accept and use them.

Undoubtedly, passenger anxiety in autonomous vehicles remains higher than that of conventional vehicles at present. Kyriakidis et al. [46] reported that 22% of 5000 respondents did not want to pay for a fully self-driving system based on current technologies. Nonetheless, nearly 70% believed that driverless cars would comprise half of the vehicles on public roads by 2050. Passenger anxiety and behavioral tendencies are formed by human experiences, values, and social norms and they clearly influence behavioral choices [53]. Thus, they are expected to affect the public acceptance of specific transport policies [54].

As the theory of the diffusion theory implies, the adoption of innovative technologies depends on how we perceive ideas or products as new, innovative, advantageous, and valuable relative to existing technologies [55]. The theory suggests that an opportunity to try or observe technologies increases the chance of technology adoption and the speed of diffusion. Hence, the following hypotheses are proposed.

**Hypothesis 1a (H1a).** *Passenger attitudes towards autonomous buses positively influence their willingness to use emerging mobility services in a closed environment.*

**Hypothesis 1b (H1b).** *Passenger perceived benefits towards autonomous buses positively influence their willingness to use emerging mobility services in a closed environment.*

**Hypothesis 1c (H1c).** *Passenger anxiety towards autonomous buses negatively influences their willingness to use emerging mobility services in a closed environment.*

**Hypothesis 1d (H1d).** *Passenger early technology adoption positively influences their willingness to use emerging mobility services in a closed environment.*

The theory suggests that attitude mediates the relationship between outcome benefits and behavioral intentions, thus affecting the motivation factors that influence behaviors. Wintersberger et al. [56] conducted a pilot study to examine passenger acceptance of self-driving cars as a regular transit mode in Australia. According to Ring [57], people who used self-driving vehicles were mostly concerned about security threats [57,58]. Parents are less willing to have their children use a driverless school bus than a conventional driver-operated bus [59].

Despite the high levels of optimistic scenarios and increasing research interests, safety concerns, and lack of trust continue to constrain customer willingness to invest in autonomous transit systems [60]. Passenger anxiety toward the overall autonomous vehicle concept is negatively associated with the benefits consumers perceive from automated technologies. However, gradual exposure to semi-autonomous vehicles in closed, low-risk environments should play a key role in reducing anxiety [36]. According to the AAA [36], the piecemeal addition of automated vehicle technologies (e.g., steering, cruise control, emergency braking, and self-parking) will help reduce and will open the door for greater acceptance. Anxiety is found to be influenced by risk perception [61].

Choi et al. [62] studied how consumers' perceived risks and benefits affected their attitudes toward their behavioral intentions. Their study found that perceived benefits positively affected attitudes, and perceived risks inversely affected consumer attitudes. Their research also found that consumer attitude was associated with perceived benefits, perceived risk, and behavioral intention. We establish the following hypotheses.

**Hypothesis 2a (H2a).** *Passenger perceived benefits towards autonomous buses mediate negatively the negative relationship between anxiety and willingness to use emerging mobility services in a closed environment.*

**Hypothesis 2b (H2b).** *Passenger anxiety towards autonomous buses mediates positively the negative relationship between perceived risks and willingness to use emerging mobility services in a closed environment.*

**Hypothesis 2c (H2c).** *Passenger anxiety towards autonomous buses mediates positively the negative relationship between perceived risks and perceived benefits in a closed environment.*

**Hypothesis 2d (H2d).** *Passenger attitudes towards autonomous buses mediate negatively the negative relationship between anxiety and willingness to use emerging mobility services in a closed environment.*

AI technology has an essential role in complementing the use of autonomous cars. Lombardo published comments by Dara Khosrowshahi, Chief Executive at Uber Technologies, Inc., that indicated that self-driving vehicles are in a learning curve [63]. However, industries already use AI technologies daily. Banks use them to detect fraud; investment firms use them to predict changes in the stock markets; insurance companies use them to produce policy quotes; hospitals use them to identify and stop diseases; human resources organizations use them to make hiring decisions; YouTube and Amazon-like companies use them to suggest consumer purchasing options (e.g., novels, videos, household products, etc.); and autonomous shuttle makers use them to navigate autonomous vehicles and monitor driver and passenger behaviors. Passenger attitude and acceptance of new, AI-based technologies are, therefore, necessary for the successful diffusion of ABs [64].

Hardman et al. [65] studied the attitudes of early technology adopters and found that they had positive perceptions of autonomous transportation, and, according to Ruggeri et al. [66], the adoption of driverless cars is a result of the adoption patterns of previous technologies. It is thus reasonable to theorize that early technology adopters will support the use of the ABs [67]. The following hypotheses are suggested.

**Hypothesis 3a (H3a).** *Passenger attitudes towards autonomous buses mediate positively between their perception of AI technologies and willingness to use emerging mobility services in a closed environment.*

**Hypothesis 3b (H3b).** *Passenger attitudes towards autonomous buses mediate positively between early technology adoption and willingness to use emerging mobility services in a closed environment.*

**Hypothesis 3c (H3c).** *Passenger early technology adoption mediates positively the relationship between their perception of AI technologies and their attitudes towards autonomous buses.*

**Hypothesis 3d (H3d).** *Passenger early technology adoption mediates positively the relationship between their perception of AI technologies and willingness to use emerging mobility services in a closed environment.*

**Hypothesis 3e (H3e).** *Passenger perceived benefits mediate positively between perception in AI technologies and willingness to use emerging mobility services in a closed environment.*

Considering that autonomous-driving is still in its early stages of development, passengers may be more forgiving and patient of related mechanical imperfections of emerging technologies, including slow acceleration, frequent braking, and cautious steering [68,69]. Xu et al. [39] found that the experience of Level-3 automated vehicles, wherein the autonomous car monitors the driving environment, and a driver is on board to take over at any time, influenced passenger trust, perceived usefulness, and user-friendliness. As passengers become more accustomed to autonomous operation, their changing attitude toward self-driving ABs is likely to improve [70]. Therefore, the following hypothesis is set.

**Hypothesis 4 (H4).** *Passengers' willingness to use emerging mobility services in a closed environment after a sample riding experience is higher than before having a sample riding experience.*

### 3. Methods

A structural equation model (SEM) was built to test the first three sets of hypotheses listed above. The SEM method is preferred for research problems like ours, because it estimates the multiple and interrelated dependencies of an integrated analysis. A series of two surveys (i.e., Surveys 1 and 2) were administrated in February–March, 2019, and April–May, 2019, respectively. Survey 1 was the pretest survey collected prior to participants having the opportunity to experience a prototype semi-autonomous Olli bus service offered by Local Motors at California State University, Sacramento. Survey 2 was the post-test survey. The same group of 146 participants answered both surveys. This section reports the results of these preference surveys about prospective passengers' willingness to ride and their concerns about ABs in Sacramento, a typical metropolitan region in the State of California, USA. As autonomous technologies progress, ABs may offer substantial efficiency and safety advantages over conventional public-transit services. However, unfamiliarity with autonomous technologies may defy its acceptance by the general public and slow the adoption process. Using the SEM modeling framework, this research examined which types of transit passengers were most willing to ride ABs.

Hypothesis H4 is tested using a nonparametric matched-pairs sign test [71]. The purpose of this research is to find determinants of willingness to use emerging mobility services in a closed environment, and to examine how the experience of riding ABs influences passengers' desires to use emerging mobility services.

Four different demographic variables towards defining the profile of participants were considered, including gender, age, education, and household income (for the year 2018). Of respondents, 56% in this study were males, and the remaining (44%) were females. Concerning age, the sample stated a higher share of respondents between 18 and 22 years old (almost 50%), whereas the other ~50% of the respondents were above 22 years old. Of respondents, 69% had at least one higher education degree (associate's or higher). Moreover, among the respondents, 18.6% earned a household income for the year 2018 less than USD 20,000, 16.6% earned between USD 20,000 to 34,999, 14.5% earned between USD 35,000 to 49,999, 22.8% earned between USD 50,000 to 74,999, and 27.64% earned more than USD 75,000.

The Olli vehicles in this study were ABs, but a trained individual was on board to reassure passengers of the vehicle's safety. The bus comprised 3D-printed parts equipped with light detection and ranging, which used invisible laser beams to build a view of the surrounding environment. It also used the Global Positioning System satellite constellation for localization.

#### 3.1. Dependent Endogenous Latent Variable: Ride Intention

Ride intention is defined as the individual assessment of future willingness to ride [70]. Our survey consisted of two questions on ride intention: "I support a driverless bus to transport people on public and private facilities," and "I am willing to use a driverless bus service on public and private facilities." Respondents indicated their intentions for the two questions using a five-point Likert scale with anchors of "strongly disagree" (1) and "strongly agree" (5). The ride intention index was derived from the weighted average score of the two questions and was treated as the dependent variable.

#### 3.2. Independent Latent Variables: Rider Perceptions

Respondents' perceptions about ABs were measured by their responses to 12 questions using a five-point Likert scale. The 12 questions covered a number of constructs: perceived benefits of ABs for the community, anxieties about riding ABs, attitudes towards ABs, perceived risks, perception of AI technologies, and new technology adoption. These constructs were selected based on a literature review [2,58,72]. The statements were arranged in an order that was determined to reduce response bias. Table 1 summarizes the variables used for this study.

**Table 1.** Research Measurements.

| Latent Constructs | Measurement Components (Five-Point Likert Scale) |
|---|---|
| Anxiety (ANX) | I feel anxious about using the driverless bus.<br>I hesitate to use the driverless bus.<br>The driverless bus is somewhat intimidating to me. |
| Perceived Risk (PR) | The quality of a driverless bus is not good enough so that I have a concern about its safety risks.<br>A driverless bus does not perform and function well enough so that I have a concern about its security risks.<br>I have reliability concerns for a driverless bus. |
| Perceived Benefits (PB) | A driverless bus has the potential to reduce accidents.<br>A driverless bus has the potential to reduce traffic congestion. |
| Attitude (ATT) | Experience with the driverless bus is fun.<br>I like to experience the driverless bus |
| Artificial Intelligence (AI) | I think AI has started to impact some works.<br>I believe AI has started to reshape our lives. |
| Early Technology Adopter (TECH) | I am likely to be more accepting of the new technology.<br>I tend to embrace new technology before most other people do. |
| Willingness to Ride (RIDE) | I support a driverless bus to transport people on the public and/or private facilities.<br>I am willing to use a driverless bus service on the public and/or private facilities. |

For Hypothesis H4, the nonparametric matched-pairs sign test for equality fit well for the purpose, because the variables included ordinal and matched pairs, and the symmetry of the distribution of the differences cannot be assumed [71].

## 4. Results

The structural equation modeling is applied to analyze the collected data. To examine the measurement model, we followed the two-step approach recommended by Anderson and Gerbing [73]. First, we conducted a confirmatory factor analysis (CFA) to measure convergent and discriminant validity. We then, through the structural model, examined the model fit and tested the hypotheses. Both the measurement model and the structural model are assessed by the maximum likelihood parameter estimator AMOS version 27 and verified using STATA version 16.

### 4.1. Measurement Model

The Cronbach's $\alpha$ scores of each construct (measurement models in both before and after ride experiences) were above 0.70, indicating that the scale used in this study has adequate internal reliability [74]. Factor loadings for the model exceeded the recommended value of 0.60 for the measurement model [75,76]. A factor loading for one of the constructs (AI) for "After-ride experience" is greater than one (1.0). However, factor loadings are coefficients, not correlations and they can be greater than one [77]. The Average Variance Extracted (AVE) values for both before and after-ride models are greater than 0.50 [78]. The results in Table 2 show the measurement model to meet the requirement of convergent validity.

To demonstrate the discriminant validity, the AVE for each factor or latent reflective construct (average variance shared between the construct and its indicators) must be greater than its shared variance with any other factor or construct, as recommended by Fornell and Larcker [78]. Table 3 shows the correlation between the construct. The measurement model (for both before and after ride experiences) satisfies the requirement of discriminant validity. The results, therefore, confirm that the instrument has satisfactory construct validity.

### 4.2. Structural Model: Structural Equation Analysis

After confirming the adequate fit for the measurement model, we evaluated the structural model and tested the research hypotheses using AMOS version 27. Figure 1 illustrates our hypothesized

model. The first step in evaluating the structural model was to compute the $R^2$ statistic, which indicates the amount of variance of the dependent variable explained by the predictor model variables. As one observes in Table 4, the value of $R^2$ for the model's dependent variable (i.e., willingness to buy) exceeds the reference value of 0.1 [79] for before ($R^2 = 0.64$) and after ($R^2 = 0.58$) ride experiences. Considering the size of the sample size, we used Comparative Fit Index (CFI), Incremental Fit Index (IFI), and Root Mean Square Error of Approximation (RMSEA) to examine the model fit. These fit indices satisfied the recommended values (CFI: 0.933 for before-ride, 0.940 for after-ride; IFI: 0.935 for before-ride, 0.941 for after-ride; RMSEA 0.069 for before-ride, 0.068 for after-ride). With evidence of acceptable fit, we proceeded to test the hypotheses.

**Table 2.** Convergent validity for before- and after-ride.

| Construct/Indicator | Item | Factor Loading | AVE | Cronbach's $\alpha$ |
|---|---|---|---|---|
| Anxiety (ANX) | ANX_03 | **0.72** (0.76) | **0.60** (0.69) | **0.82** (0.87) |
| | ANX_02 | **0.90** (0.87) | | |
| | ANX_01 | **0.69** (0.87) | | |
| Perceived risk (PR) | PR_03 | **0.76** (0.65) | **0.65** (0.54) | **0.84** (0.77) |
| | PR_02 | **0.86** (0.82) | | |
| | PR_01 | **0.79** (0.73) | | |
| Perceived benefit (PB) | PB_03 | **0.98** (0.96) | **0.65** (0.66) | **0.72** (0.75) |
| | PB_02 | **0.58** (0.63) | | |
| Attitude (ATT) | ATT_03 | **0.99** (0.91) | **0.63** (0.76) | **0.68** (0.87) |
| | ATT_02 | **0.53** (0.84) | | |
| Artificial Intelligence (AI) | AI_02 | **0.81** (0.57) | **0.62** (0.74) | **0.76** (0.76) |
| | AI_01 | **0.76** (1.07) | | |
| Early technology adopter (TECH) | TECH_02 | **0.67** (0.71) | **0.65** (0.55) | **0.75** (0.70) |
| | TECH_01 | **0.92** (0.78) | | |
| Willingness to ride (RIDE) | RIDE_02 | **0.85** (0.84) | **0.74** (0.74) | **0.85** (0.84) |
| | RIDE_01 | **0.87** (0.86) | | |

Note: The figure in bold is "Before-ride experience". The figure in parenthesis is "After-ride experience."

**Table 3.** Correlation between the construct for before- and after-ride.

|       | ANX    | PR     | PB     | ATT    | AI     | TECH   | RIDE   |
|-------|--------|--------|--------|--------|--------|--------|--------|
| ANX   | **0.60** (0.69) |        |        |        |        |        |        |
| PR    | 0.45 (0.35) | **0.65** (0.54) |        |        |        |        |        |
| PB    | 0.15 (0.15) | 0.14 (0.11) | **0.65** (0.66) |        |        |        |        |
| ATT   | 0.16 (0.16) | 0.06 (0.15) | 0.02 (0.06) | **0.63** (0.76) |        |        |        |
| AI    | 0.00 (0.01) | 0.07 (0.05) | 0.06 (0.01) | 0.11 (0.15) | **0.62** (0.74) |        |        |
| TECH  | 0.02 (0.15) | 0.01 (0.06) | 0.01 (0.03) | 0.09 (0.38) | 0.12 (0.06) | **0.65** (0.55) |        |
| RIDE  | 0.31 (0.35) | 0.21 (0.25) | 0.17 (0.15) | 0.43 (0.41) | 0.14 (0.06) | 0.15 (0.43) | **0.74** (0.74) |

Note: Diagonal elements in bold (e.g., 0.60 for ANX × ANX) are AVEs for "Before-ride experience." Figures in parentheses (e.g., 0.69 for ANX × ANX) in diagonal elements are AVEs for "After-ride experience." Other figures are square of the correlation (e.g., 0.45 for ANX × PR) between the construct for "Before-ride experience." Figures in parentheses are square of the correlation (e.g., 0.35 for ANX × PR) between the construct for "After-ride experience".

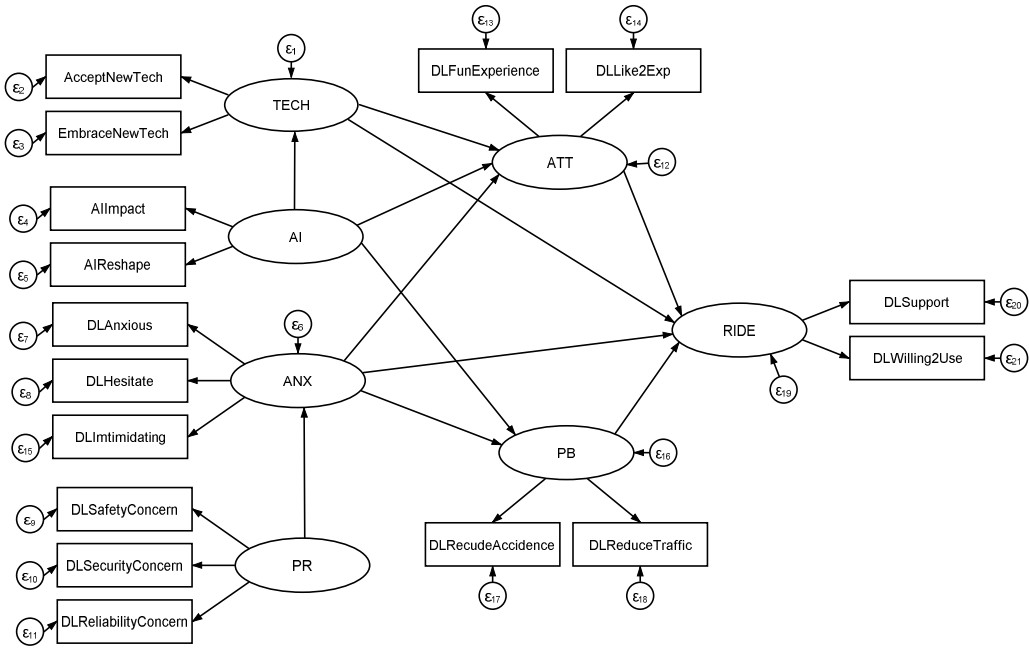

**Figure 1.** Hypothesized SEM.

Hypotheses H1a stated consumer's attitude positively influences the willingness to ride ABs. Our results supported H1a (Before: β = 0.73, t = 5.13; After: β = 0.27, t = 2.16). Hypotheses H1b examined if consumer's perceived benefit positively influences the willingness to ride ABs. We received consistent results for before-ride (β = 0.22, t = 2.59) and after-ride (β = 0.13, t = 1.67). Therefore, H1b was supported. Hypotheses H1c stated consumers' anxiety negatively influences the willingness to ride ABs. Our results supported H1c (Before: β = −0.17, t = −1.77; β = −0.33, t = −4.36). Hypotheses H1d stated consumers' technology adoption positively influences the willingness to ride ABs. Our results supported this hypothesis (Before: β = 0.24, t = 2.36; After: β = 0.46, t = 3.09). Table 4 summarizes the results of the main effect analysis. The parameters were estimated using a group-based maximum likelihood method where responses with missing values in the after-ride group were removed.

**Table 4.** The results of the main effect analysis.

| | Coef. | Std. Err. | T | *p*-Value | [95% | C.I.] | |
|---|---|---|---|---|---|---|---|
| **RIDE** | | | | | | | |
| **←ATT** | | | | | | | |
| Before | 0.734 | 0.143 | 5.13 | 0.000 *** | 0.454 | 1.015 | H1a |
| After | 0.273 | 0.127 | 2.16 | 0.031 ** | 0.025 | 0.521 | Supported |
| **←PB** | | | | | | | |
| Before | 0.218 | 0.084 | 2.59 | 0.010 ** | 0.053 | 0.383 | H1b |
| After | 0.125 | 0.075 | 1.67 | 0.096 * | −0.022 | 0.273 | Supported |
| **←ANX** | | | | | | | |
| Before | −0.173 | 0.097 | −1.77 | 0.076 * | −0.364 | 0.018 | H1c |
| After | −0.333 | 0.076 | −4.36 | 0.000 *** | −0.483 | −0.183 | Supported |
| **←TECH** | | | | | | | |
| Before | 0.241 | 0.102 | 2.36 | 0.018 * | 0.041 | 0.441 | H1d |
| After | 0.463 | 0.150 | 3.09 | 0.002 *** | 0.169 | 0.756 | Supported |

*** significant at <0.01, ** significant at <0.05, * significant at <0.1.

The Cronbach's $\alpha$ scores of each construct (measurement models in both before and after ride experiences All figures and tables should be cited in the main text as Figure 1, Table 1, etc.

To test the effect of a third valuable or mediating variable, we used the bootstrap confidence intervals test method in AMOS version 27. Table 5 shows the results of the mediating effects in this study. A bootstrap analysis with 200 resamples indicated indirect effect for anxiety on the willingness to ride ABs via perceived benefits (H2a supported, $p = 0.03$ for before-ride and $p = 0.02$ for after-ride); indirect effect for perceived risks on the willingness to ride ABs via anxiety (H2b generally supported, $p = 0.08$ for before-ride, $p = 0.01$ for after-ride); indirect effect for perceived risks on perceived benefits via anxiety (H2c supported, $p = 0.001$ for before-ride and $p = 0.01$ for after-ride); indirect effect for anxiety on the willingness to ride ABs via attitudes (H2d supported, $p = 0.01$ for before-ride and after-ride); indirect effect for AI on the willingness to ride ABs via attitudes (H3a supported, $p = 0.03$ for before-ride and $p = 0.04$ for after-ride); indirect effect for AI on attitudes via early technology adaption (H3c supported, $p = 0.05$ for before-ride and $p = 0.02$ for after-ride); and indirect effect for AI on attitudes on the willingness to ride ABs via early technology adaptation (H3d supported, $p = 0.03$ for before-ride and $p = 0.01$ for after-ride). However, this study found mixed results for the indirect effect for early technology adoption on the willingness to ride ABs via attitudes (H3b supported for after-ride, but not before-ride), and the indirect effect for AI on the willingness to ride ABs via early technology adaptation (H3e supported for before-ride, but not after-ride).

For H4, the nonparametric matched-pairs sign test compared the two pairs of endogenous observations:

- "I support a driverless bus to transport people on public and/or private facilities" Before vs. After
- "I am willing to use a driverless bus service on public and/or private facilities" Before vs. After

The test results are summarized in Table 6. The following hypothesis was significantly supported by data.

- H4: Passengers' willingness to use emerging mobility services in a closed environment after a sample riding experience is higher than before having a sample riding experience.

**Table 5.** The results of the mediating effect analysis.

| Parameter | Before | | | | After | | | | Hypothesis |
|---|---|---|---|---|---|---|---|---|---|
| | Estimate | Lower | Upper | *p* | Estimate | Lower | Upper | *p* | |
| **ANX-PB-RIDE** | −0.09 | −0.18 | −0.02 | 0.03 | −0.06 | −0.17 | −0.01 | 0.02 | H2a: Supported |
| **PR-ANX-RIDE** | −0.16 | −0.35 | −0.01 | 0.08 | −0.23 | −0.44 | −0.11 | 0.01 | H2b: Supported |
| **PR-ANX-PB** | −0.34 | −0.55 | −0.22 | 0.00 | −0.32 | −0.50 | −0.16 | 0.01 | H2c: Supported |
| **ANX-ATT-RIDE** | −0.20 | −0.40 | −0.09 | 0.01 | −0.06 | −0.20 | −0.02 | 0.01 | H2d: Supported |
| **AI-ATT-RIDE** | 0.16 | 0.02 | 0.33 | 0.03 | 0.09 | 0.01 | 0.26 | 0.04 | H3a: Supported |
| **TECH-ATT-RIDE** | 0.10 | −0.00 | 0.30 | 0.11 | 0.13 | 0.02 | 0.27 | 0.04 | H3b: After-ride supported |
| **AI-TECH-ATT** | 0.07 | 0.01 | 0.28 | 0.05 | 0.19 | 0.06 | 0.36 | 0.02 | H3c: Supported |
| **AI-TECH-RIDE** | 0.07 | 0.01 | 0.25 | 0.03 | 0.13 | 0.04 | 0.32 | 0.01 | H3d: Supported |
| **AI-PB-RIDE** | 0.06 | 0.01 | 0.17 | 0.01 | 0.02 | −0.01 | 0.08 | 0.21 | H3e: Before-ride supported |

**Table 6.** Comparison of passengers' willingness to use autonomous buses (ABs): before-ride and after-ride.

| Sign | Observed | Expected | Sign | Observed | Expected |
|---|---|---|---|---|---|
| positive | 19 | 44 | positive | 26 | 42 |
| negative | 69 | 44 | negative | 58 | 42 |
| zero | 47 | 47 | zero | 51 | 51 |
| all | 135 * | 135 * | all | 135 * | 135 * |

One-sided tests:
  Ho: median of DLSupport1—DLSupport2 = 0 vs.
  Ha: median of DLSupport1—DLSupport2 > 0
    Pr (#positive ≥ 19) =
      Binomial ($n$ = 88, x ≥ 19, $p$ = 0.5) = 1.0000
  Ho: median of DLSupport1—DLSupport2 = 0 vs.
  Ha: median of DLSupport1—DLSupport2 < 0
    Pr (#negative ≥ 69) =
      Binomial ($n$ = 88, x ≥ 69, $p$ = 0.5) = 0.0000
Two-sided test:
  Ho: median of DLSupport1—DLSupport2 = 0 vs.
  Ha: median of DLSupport1—DLSupport2 != 0
    Pr (#positive ≥ 69 or #negative ≥ 69) =

One-sided tests:
  Ho: median of DLWilling2Use1—DLWilling2Use2 = 0 vs.
  Ha: median of DLWilling2Use1—DLWilling2Use2 > 0
    Pr (#positive ≥ 26) =
      Binomial (n = 84, x ≥ 26, p = 0.5) = 0.9999
  Ho: median of DLWilling2Use1—DLWilling2Use2 = 0 vs.
  Ha: median of DLWilling2Use1—DLWilling2Use2 < 0
    Pr (#negative ≥ 58) =
      Binomial ($n$ = 84, x ≥ 58, p = 0.5) = 0.0003
Two-sided test:
  Ho: median of DLWilling2Use1—DLWilling2Use2 = 0 vs.
  Ha: median of DLWilling2Use1—DLWilling2Use2 != 0
    Pr (#positive ≥ 58 or #negative ≥ 58) =

\* Incomplete responses were removed.

A series of sign tests (Table 7) were conducted to compare all pairs of variables before- and after the ride experience. Insignificant differences were observed for AI and TECH measures that make sense. The passengers perceived stronger traffic reduction after-ride. They also had significantly fewer risk concerns, as well as significantly lower anxiety, after the ride experience. These contributed to their significantly increased attitude towards and willingness to use ABs after-ride.

**Table 7.** Before-ride and after-ride comparison: a summary.

|      | Ho | *p* | Ha |
|------|----|-----|-----|
| TECH | AcceptNewTech1-AcceptNewTech2 = 0 | 0.263 | insignificant |
|      | EmbraceNewTech1-EmbraceNewTech2 = 0 | 0.846 | insignificant |
| AI   | AIImpact1-AIImpact2 = 0 | 0.651 | insignificant |
|      | AIReshape1-AIReshape2 = 0 | 0.780 | insignificant |
| PB   | DLRecudeAccidence1-DLRecudeAccidence2 = 0 | 0.156 | insignificant |
|      | DLReduceTraffic1-DLReduceTraffic2 = 0 | 0.016 | DLReduceTraffic1-DLReduceTraffic2 < 0 |
| PR   | DLSafetyConcern1-DLSafetyConcern2 = 0 | 0.000 | DLSafetyConcern1-DLSafetyConcern2 > 0 |
|      | DLSecurityConcern1-DLSecurityConcern2 = 0 | 0.000 | DLSecurityConcern1-DLSecurityConcern2 > 0 |
|      | DLReliabilityConcern1-DLReliabilityConcern2 = 0 | 0.000 | DLReliabilityConcern1-DLReliabilityConcern2 > 0 |
| ANX  | DLAnxious1-DLAnxious2 = 0 | 0.000 | DLAnxious1-DLAnxious2 > 0 |
|      | DLHesitate1-DLHesitate2 = 0 | 0.000 | DLHesitate1-DLHesitate2 > 0 |
|      | DLImtimidating1-DLImtimidating2 = 0 | 0.000 | DLImtimidating1-DLImtimidating2 > 0 |
| ATT  | DLFunExperience1-DLFunExperience2 = 0 | 0.000 | DLFunExperience1-DLFunExperience2 < 0 |
|      | DLLike2Exp1-DLLike2Exp2 = 0 | 0.006 | DLLike2Exp1-DLLike2Exp2 < 0 |
| RIDE | DLSupport1-DLSupport2 = 0 | 0.000 | DLSupport1-DLSupport2 < 0 |
|      | DLWilling2Use1-DLWilling2Use2 = 0 | 0.000 | DLWilling2Use1-DLWilling2Use2 < 0 |

## 5. Discussion

In this study, we developed an exploration model that applied a survey of 146 prospective passengers of emerging mobility services with the ABs operated in a closed-environment. To the best of our knowledge, this was the first pretest/posttest-designed study that investigated the underlying factors of the acceptance of semi-ABs operated in a closed environment. However, we must note the study limitation of the sample. Because participants in this study are young, educated, and tech-savvy students who use computers and the Internet widely for many of their daily activities, the results of this study should be interpreted with caution.

Passenger beliefs and general attitudes toward products are important to marketers because they affect behavior. Thus, an understanding of such factors should help the transportation industry identify interested passengers and expand markets. This study has examined (1) direct effects between passenger willingness to use emerging mobility services and anxiety, perceived benefits, and attitudes, and (2) effects of indirect or third variables, such as attitudes, perceived societal benefits, anxiety, early technology adaptation, and beliefs in AI technologies. The results indicate that while passengers both before and after ride experiences share common sentiments toward ABs, a majority of the issues examined by this study shows a dichotomy between before and after ride experiences. Our findings concur with research by Anania et al. [59]. We found that passenger attitudes, perceived benefits, and anxiety are significant predictors of willingness to use emerging mobility services both before and after ride experiences. In addition to these direct effects, this study has examined the mediation effects of attitudes, perceived benefits, anxiety, and technological variables. The examination of mediation is important for the theory of development for the consumer perception of autonomous transportations. It can allow us to investigate the key processes involved in making optimistic outcomes. Our findings show the effect of perceived risks on willingness to use emerging mobility services through anxiety. When passengers have a concern about safety and reliability concerns about AB, they will tend to feel more anxious about using emerging mobility services, thus reducing their willingness to use emerging mobility services. If passengers feel less intimidating to use emerging mobility services, they tend to perceive the benefits of emerging mobility services, thus increasing their willingness to use emerging mobility services.

The results of this study found mixed results for the two mediations. First, this study did not support the following hypothesis for the before-ride experience: passenger attitude mediates positively the relationship between early technology adaptation and willingness to use emerging mobility services in a closed environment. Early technology-adopters overlook the rough edges and potential challenges of being among the first and jump into markets early. Perhaps, before interaction with emerging

mobility services, passengers did not feel excited about the emerging mobility services. Once they experience emerging mobility services and find the experience fun, they tend to support the emerging mobility services, therefore supporting the hypothesis for the after-ride experience, but not for the before-ride. Early adopters are important for the diffusion of autonomous technologies because they represent opinion leaders. They embrace change- opportunities and are comfortable adopting new, innovative ideas. Passengers who are early technology adopters tend to perceive the semi-ABs in a positive light. This research found that those who considered themselves early adopters of new technology tended to have positive attitudes toward autonomous technologies. Therefore, they had a higher willingness to use emerging mobility services. Our findings revealed significant increases in the desire to use public transportation services based on autonomous driving following an example riding experience. The results of this study are consistent with past research that triability, or the degree to which customers can try a product or service, is highly related to technology adoption [80,81].

Second, this study did not support the following hypothesis for the direct effect between AI technology perception and perceived benefits for the after-ride experience. Consequently, the study did not observe the following mediation for the after-ride experience: Passenger perceived benefits mediate positively between perception in AI technologies and willingness to use emerging mobility services. Many participants agreed that AI's impact on society. Perhaps they could not link how AI technologies would help reduce traffic congestion and traffic accidents. Holistic passenger-AB interaction would influence their image of "AI technologies." Any negative ride-experience would influence their perception of "AI technologies." This study identified some future challenges related to self-driving ABs under a closed environment [82].

In many cases, the passenger seats in ABs are made of 3D-printed parts that result in slippery seat surfaces. In a closed-environment, there are many stop requirements because AB operations must consider random behaviors of pedestrians, bicyclists, skateboarders, and even squirrels. Some participants in this study also shared their concerns, stating

- I think the best improvement for the ABs would be smoother stops. When they finished, it was a bit abrupt, and I found myself swaying to the side a bit.
- Although the AI and the algorithms it uses are impressive, I believe that there can be improvements in braking mechanics. The bus can come at a steadier stop than a sudden stop.
- When the bus stopped, it was quite sudden and not very smooth, and turns were a bit awkward.

Thus, when the ABs unexpectedly braked, passengers slide around on their seats. Perhaps concerns expressed by passengers in this study partially explain support for the relationship between anxiety and willingness to ride. In our interview, a manager of an AB manufacturer said: "In a closed, low-speed environment, people assume it is easy to develop ABs. It is not." The manager who served as a technician/safety officer was always on board to assist and answer questions of passengers and to take over at any time. This perhaps explains the relationship between risk perception and willingness to ride even after-ride experience. In February 2020, a woman on an AB operating in Columbus, Ohio, slid out of her seat onto the floor. Soon after, the NHTSA announced that it would not permit driverless vehicles to operate with passengers anywhere in the US until a re-confirmation of safety and technical concerns was accomplished [83]. Gaining trust from potential customers regarding safety is critical to the overall development and deployment of ABs [46].

The results of this study provide evidence that interactions with transportation employing emerging technologies increase willingness to use emerging mobility services operated by AI technologies as the experience can help reduce their risk concerns and anxiety. Our findings are consistent with autonomous vehicle research in consumer perception. Penmesta et al. [84] found interactions with autonomous vehicles help passengers increase their safety perceptions and approve AI technologies.

The recent worldwide pandemic caused nations to recognize the benefits of emerging mobility services. Many people probably agree that self-driving transportation is useful in the fight against

the pandemic, because it reduces the burden of those engaged in the transportation of patients or medical supplies or food deliveries, especially in infected areas or hospitals or in situations where social distancing policies are in place.

ABs provide transportation options for simpler, slower, and relatively controlled environments. They complement the current lack of autonomous vehicles in such areas as university campuses, senior-citizen complexes, and medical facilities. As we learn more about AI-based technologies, our perceptions and behavioral responses are updated to some extent. More work in this area is required globally. Generally, attitudes lead people to behave in a consistent way toward their choices of transit, because they are not required to interpret and freshly react to every decision. Thus, attitudes economize the energy and thoughts of our decision-making processes.

Manufacturers employing emerging technologies face constant technological changes with the continual shifting of the competitive landscape [85]. Substantial components of the ABs are modular, which enable system component exchange. This modular architecture results in quicker product development, provides for better adaptation to different customer segments, and facilitates specific types of innovation. The modular aspect also leads to greater imitation with the undermining of the sustained market performance of a firm [86]. Therefore, it is difficult for the manufacturers to differentiate ABs, because competitors easily imitate technology. Therefore, AB manufacturers should carefully examine passenger opinions to help them effectively respond to needs.

This study finds that a pleasant riding experience is a vital influencing factor toward a willingness to use emerging mobility services. The research conducted by Solbraa Bay [87] also reported that attitude is the most significant predicting factor of passenger's willingness to drive autonomous cars. Passengers need to know what ABs do and how they behave in real driving situations. Passengers will use ABs if they can establish trust with the technology-based transit system by clearing issues of safety and security. The potential benefits of partial or full ABs will not be realized without public acceptance, particularly from regular bus passengers.

## 6. Conclusions

The results of this study indicate that a combination of factors (e.g., benefits, risks, anxieties, attitudes, and technology adoption) directly or indirectly influence an individual's acceptance of ABs. Generally, the current study reveals that ABs are perceived as an acceptable form of community transportation as long as passengers recognize the benefits, have a positive attitude, and feel less anxious. The autonomous transportation industry must pay attention to studies such as these that track consumer perception about safety and trust. A survey conducted by a major national association with over 620-million members showed that respondents were particularly cautious about autonomous vehicles and that only 12% would trust self-driving cars [36]. Thus, if passengers believe that ABs are more beneficial at the individual and environmental levels and after they have enjoyable ride experiences, they will show a more positive attitude towards the semi-ABs. In turn, their willingness to use the emerging mobility services will increase.

Manufacturers of driverless vehicles and governments have many challenges to overcome prior to making the emerging mobility services commercially available. Accidents must be prevented, emissions must be reduced, and energy efficiency and road infrastructure must be improved. These and other factors underlie the acceptance of autonomous technologies on dedicated roads under closed environments [20]. Policymakers must establish laws aimed at protecting customers and the natural environment. Manufacturers of transportation employing emerging technologies can become first movers in mobility services if they can turn policymakers into allies by going above and beyond existing laws.

Many reports on the impact of economic factors and governance on gas emissions (e.g., [88]; Special Issue "Electric Vehicles: New Challenges and Opportunities for Sustainability" in Sustainability). According to the US Environmental Protection Agency [89], the share of US emissions from the transportation sector accounted for 29% in 2017. The share of emission gases produced by buses

is small, compared with passenger cars. However, the number of emissions produced by buses increased by 140% between 1990 and 2017. To achieve the global diffusion of ABs and thus reach the new environmental potential, broad acceptance of new technology-based mobility concepts must be achieved [90].

Passengers' attitudes are difficult to change. After their attitudes settle into a consistent pattern, changing a single person might require significant adjustments. Thus, AB manufacturers and marketers are well-advised to consider developing safe, secure, and comfortable ABs and operating them along user-friendly routes. When passengers experience the ride, especially for the first time, they will develop overall positive attitudes toward ABs. A safety officer could be on board to explain the features and mechanisms of the vehicles during the ride to remove or reduce anxiety. An article in Forbes entitled "Why Self-Driving Cars Might Never Become A Commodity" [91] provided legitimate concerns about autonomous technologies and discussed uncontrolled vs. controlled fixed environments. Nonetheless, appropriate attention must be paid to passenger expectations.

**Author Contributions:** Conceptualization, K.C. and M.M.; methodology, K.C. and Y.S.; software, Y.S.; validation, K.C. and Y.S.; formal analysis, K.C. and Y.S.; investigation, K.C. and Y.-Y.C.; data curation, K.C.; writing—original draft preparation, K.C., Y.S. and Y.-Y.C.; writing—review and editing, M.M.; funding acquisition, M.M. All authors have read and agreed to the published version of the manuscript.

**Funding:** This research was funded by the Environmental Restoration and Conservation Agency (ERCA) of Japan, grant number JPMEERF16S11603 of the Environmental Research and Technology Development Fund.

**Conflicts of Interest:** The authors declare no conflict of interest.

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
