# Peer review of "Towards a Sustainable Society through Emerging Mobility Services: A Case of Autonomous Buses"

_sustainability, doi:10.3390/su12219170_

Round 1

Reviewer 1 Report

The article, entitled: Towards a Sustainable Society Through Emerging Mobility Services: A Case of Autonomous Buses, touches on an interesting and up-to date topic. The paper presents a well-designed research project. The experiments together with pretest-and-posttest methods are valuable for understanding the users behaviors under the specific conditions.

Aim, hypothesis are clearly stated. Method used is explained and results showed, however I have some remarks to the composition of the paper.

Section 2: Conceptual Background… Rows: 155-170 The first paragraph is a continuation or even a repetition of what was in the Introduction section. These considerations should be in the Introduction section. Likewise rows: 178-185. In the section devoted to hypotheses, the benefits of ABs should not be discussed again, but the users' behavior issues and attitudes, as this is the subject of research.

In the discussion section, the authors should refer to the fact that their study group was quite specific. They were young, educated people who are definitely more tech-savvy and more open to new things.

A large part of the Conclusion section is actually a further discussion of the authors about the benefits and the future of autonomous vehicles. The Conclusion should only include a summary of the results obtained, which concerned the attitudes of users, and not the benefits of using autonomous technology. Thus, in the Conclusion section, new footnotes are rather not introduced, because the Conclusion is only a paraphrase and a summary of the authors' conclusions. Subsection 6.1.1 are the conclusions of the study. Subsection 6.1.2. should be in the discussion. Section 6.2 as well, follows the discussion about the future of ABs.

Please note also that, changing the layout of the article, the authors should pay attention to eliminate the repetition in the content.

Technical remarks:

  • Row 99: First time used the abbreviation “AAA” – please write whole name

Reviewer 2 Report

This paper presents a social study for the adoption of autonomous mobility services, in particular autonomous buses.

In particular, I think autonomous vehicles will be working in the future, but I don't think autonomous buses are the best option.
Actually, I think that, if there are robo taxis, autonomous buses are not necessary, because smaller vehicles can calculate a more optimal route for the passengers than an autonomous bus that don't have this posibility.

In addition to this, I think it is easier to have an autonomous car than an autonomous bus due to the size of the vehicle an the data that cars send to the cloud nowadays.

Regarding the social study of the paper, it is interesting the perception of the people at this moment with this incipient technology. I am not expert in the sociological study of this aspect.

Round 2

Reviewer 2 Report

I think there are better ideas for autonomous vehicles than autonomous buses.

Author Response

Thank you for your comment. We agree that the advantages of autonomous buses over other type autonomous vehicles in terms of overall energy saving effects are open to debate. However, we believe that passengers’ perception toward autonomous buses has commonalities with their perception to other type autonomous vehicles, and thus, we believe this study’s results can help us understand passengers’ perception toward autonomous vehicles.